# Atmospheric Aerosol Distribution in 2016–2017 over the Eastern European Region Based on the GEOS-Chem Model

**Gennadi Milinevsky [1,2,3,*]** , **Natallia Miatselskaya [4]** , **Asen Grytsai [2]** , **Vassyl Danylevsky [5]** , **Andrey Bril [4]** , **Anatoli Chaikovsky [4]** , **Yulia Yukhymchuk [3]** , **Yuke Wang [1]** , **Anatoliy Liptuga [6]** , **Volodymyr Kyslyi [6]** , **Olena Turos [7]** and **Yuriy Serozhkin [6]**

1   International Center of Future Science, College of Physics, Jilin University, Changchun 130012, China; wangyk16@mails.jlu.edu.cn
2   Physics Faculty, Taras Shevchenko National University of Kyiv, Kyiv 01601, Ukraine; a.grytsai@gmail.com
3   Department for Atmospheric Optics and Instrumentation, Main Astronomical Observatory, Kyiv 03143, Ukraine; juliyuhim@gmail.com
4   Institute of Physics of the National Academy of Sciences of Belarus, Minsk 220072, Belarus; nata.miat@gmail.com (N.M.); andrey.bril@gmail.com (A.B.); anatoli.chaikovsky@gmail.com (A.C.)
5   Astronomical Observatory, Taras Shevchenko National University of Kyiv, Kyiv 04053, Ukraine; vdanylevsky@gmail.com
6   V. Lashkaryov Institute of Semiconductor Physics of the National Academy of Sciences of Ukraine, Kyiv 03028, Ukraine; lipantt@gmail.com (A.L.); kyslyij@gmail.com (V.K.); yuseroz@gmail.com (Y.S.)
7   Laboratory of Air Quality, Marzeiev Institute for Public Health, National Academy of Medical Sciences of Ukraine, Kyiv 02660, Ukraine; eturos@gmail.com
*   Correspondence: genmilinevsky@gmail.com or gmilin@univ.kiev.ua; Tel.: +38-050-3525498

**Abstract:** The spatial and temporal distributions of atmospheric aerosols have been simulated using the GEOS-Chem model over the sparsely investigated Eastern European region. The spatial distribution of the particulate matter ($PM_{2.5}$) concentration, mineral dust, black carbon, organic aerosols, sea salt, as well as nitrate, sulfate, and ammonium aerosols during 2016–2017 were considered. The aerosols' concentration, seasonality and spatial features were determined for the region. Particulate matter ($PM_{2.5}$) contamination prevails in Poland in late autumn and winter. The monthly mean $PM_{2.5}$ concentration reached 55 µg m$^{-3}$ over the Moscow region in the early spring of both years. The mineral dust concentration varied significantly, reaching 40 µg m$^{-3}$ over the southwestern part of Eastern Europe in March 2016. The areas most polluted by black carbon aerosols were the central and southern parts of Poland in the winter. The organic aerosols' concentration was the largest in March and April, reaching 10 µg m$^{-3}$ over East Belarus. The sea salt aerosol concentration increased in the coastal regions in winter due to the wind strength. Mineral dust aerosols in Eastern Europe are mainly composed of dust, partially transported from the Ukrainian steppe and partially from the Saharan Desert.

**Keywords:** aerosols; GEOS-Chem; $PM_{2.5}$; mineral dust; black carbon; organic aerosol; sea salt aerosol

---

## 1. Introduction

Studies of aerosols' spatiotemporal distribution on global and regional scales are critical for climate research and air quality monitoring. The aerosol distribution and properties are determined from observational data obtained by various techniques, such as in situ measurements and ground-based and satellite remote sensing, for the purposes of determining the aerosol load and for radiative transfer

investigations (see, e.g., [1–3]). The remote sensing datasets are rather sparse in time and space, particularly because of the night and clouds. The ground-based sun-photometer measurements of the aerosol optical depth (AOD) are very accurate but limited in time and space [4,5]. The satellite remote sensing measurements of the AOD have a larger coverage area, but the confidence of the satellite data was estimated as low [2]. The spatiotemporal variability of the aerosols' concentration in the atmosphere varies with altitude because of the short lifetime of the aerosol particles in the troposphere: from hours to weeks. The content and spatiotemporal distribution of the aerosol particles, their chemical composition and microphysical structure, are defined by the physical conditions in the atmosphere and their seasonal variations [6]. The seasonal variations in the aerosol concentration and properties were detected by many observations over the globe in different conditions (e.g., [7–9]).

It was estimated from various observations that the mesoscale variability of the aerosol horizontal distribution for spatial scales is 40–400 km and for temporal scales is 2–48 h [10]. The modeling of the spatial and temporal distributions of the aerosol concentration and properties is a technique that allows for the estimation of the complete pattern of the atmospheric aerosols. These models are also an appropriate tool for climatology studies over large domains and long periods of time.

To simulate the spatiotemporal distribution of the aerosols' concentration and properties, different models are used, which are developed by different national and international institutions. For example, the Goddard Earth Observing System-5 (GEOS-5) atmospheric general circulation model (AGCM) is currently in use in the NASA Global Modeling and Assimilation Office (GMAO) at a wide range of resolutions for a variety of applications [11,12]. The widely applicable GEOS-Chem software package is a global three-dimensional (3D) model of atmospheric chemistry controlled by assimilated meteorological observations from the Goddard Earth Observing System (GEOS) of the NASA Data Assimilation Office [13,14].

The widely applicable CHIMERE model [15] is dedicated to the study of regional atmospheric pollution events. This model was applied to study the biomass-burning products' propagation and their influence on solar radiation in the atmosphere caused by intensive wildfires in the central regions of Russia during the summer of 2010 [16,17] and to study the movement of storm dust from the Sahara and biomass-burning smoke from wildfires (Heinold et al. [18]). Additionally, models are widely used to study the particulate matter ($PM_{2.5}$) distribution in the atmosphere at regional and global scales (e.g., [19–21]).

Modeling of the spatiotemporal distribution of the aerosols' concentration and their properties is particularly important for regions where up-to-date observations are sparse or absent, such as Eastern Europe. The results of the spatiotemporal distribution of the aerosols and their sources over this region using ground-based and satellite observations are discussed in [9,22,23] and simulated in [24,25]. The AOD at 865 nm were used for the analysis of aerosol properties in the atmosphere over Ukraine and adjacent territories in 2003–2011 by Bovchaliuk et al. [22], with a resolution of approximately 18 km × 18 km over cloud-free regions. These POLDER/PARASOL data describe the fine-mode aerosols from the biomass-burning products and different anthropogenic sources [26]. The increased AOD (865 nm) values have usually been observed in April–May and August–September and agree with AERONET data over the Ukraine and Moldova areas and the South of Russia. An increased AOD (865 nm) to 0.2–0.5 was observed in August 2010 due to the intensive wildfires and fire products by the transboundary transport in July–August 2010 [16,27–29].

The seasonal aerosol properties in the Eastern European region were investigated by Milinevsky et al. [9]. The seasonal variations of the aerosol amount and optical features were analyzed using the AERONET sun-photometers data and POLDER satellite measurements. These areas are influenced by the local aerosol pollution sources (traffic, industry) and by the aerosol transport from remote sources (e.g., open steppe areas, mines, and wildfires). The potential influence of air transport on the seasonal variation of aerosols based on the data from the Kyiv and Minsk AERONET sites has been discussed in [9]. The aerosol properties in the atmosphere over Eastern Europe in 2002–2019 have been investigated via MODIS data by Filonchyk et al. [30].

The global chemical transport model, GEOS-Chem, was used to simulate the spatiotemporal distributions of atmospheric concentrations of specific aerosol types over the Belarus–Ukraine region for 2010–2013 [24,25]. Results of the model simulation were evaluated using data from the Minsk and Kyiv AERONET sites. To provide a correct comparison, the simulated aerosol component concentrations were converted into coarse, fine, and total aerosol column volume concentrations, taking into account the hygroscopic growth of particles. The model-predicted and remotely sensed aerosol volume concentrations are in reasonably good agreement.

The main purpose of this paper is to determine typical aerosols and reveal the distinctive features of their spatiotemporal distributions in the atmosphere over Belarus and Ukraine where detailed in situ aerosol observations are not available. We use the GEOS-Chem model to simulate the variation of aerosols' monthly properties in the atmosphere over the Eastern European region, with a special focus on Belarus and Ukraine. In these countries, the air quality networks are not developed and currently are in the stage of planning and discussion. The aerosol distribution in the near-surface layer below 100 m has been modeled. This layer is important to study the impact of aerosols on air quality. In Section 2, the GEOS-Chem model is shortly described, and an example of the simulation of the $PM_{2.5}$ spatial distribution is presented. In Section 3, the results of the distribution of various types of aerosols over Eastern Europe according to the GEOS-Chem model are considered, followed by a discussion in Section 4. The conclusions are presented in Section 5.

## 2. Materials and Methods

GEOS-Chem, a global three-dimensional chemical transport model, is a convenient tool for evaluating the spatial distribution and transport of aerosols and other atmospheric chemical constituents [11,13]. The model was developed and used by research groups worldwide as a versatile tool, applicable to a broad range of atmospheric composition problems. General management of GEOS-Chem is provided by the team based at Harvard University and Dalhousie University and supported by the U.S. NASA Earth Sciences Division, the Canadian National Science and Engineering Research Council, and the Nanjing University of Information Sciences and Technology (http://www.geos-chem.org). The model has been successfully applied to investigate a number of tropospheric chemistry processes (see e.g., [31–34]).

The GEOS-Chem Classic configuration uses archived meteorological data from the NASA GMAO on a latitude–longitude grid to model horizontal and vertical transport (http://www.geos-chem.org/). Tracer advection in the GEOS-Chem Classic is performed using a semi-Lagrangian scheme created by Lin and Rood [35]. In this model, the dry deposition is based on the resistance-in-series method [36,37]. The scheme describes dry deposition caused by the turbulent transport in the aerodynamic layer, molecular diffusion through the boundary layer, and uptake at the surface. The aerosol dry deposition also considers the gravitational settling [38–41]. The wet deposition scheme includes the scavenging in convective updrafts, the in-cloud rainout, and the below-cloud washout [42,43]. GEOS-Chem uses the Harvard–NASA Emissions Component (HEMCO) (http://wiki.seas.harvard.edu/geos-chem/index.php/HEMCO; [44]) to calculate emissions from different sources. The HEMCO can read several types of emission inventories including anthropogenic, biomass-burning, dust, sea salt, biogenic, and others. GEOS-Chem includes the detailed tropospheric chemistry of $HO_x$-$NO_x$-VOC and ozone-halogen-aerosol.

The GEOS-Chem model allows for the study of the spatial and temporal distribution of atmospheric species. The results of model processing include the aerosol species: elemental (black) carbon, organic, mineral dust, sea salt, nitrate, sulfate, and ammonium. The aerosol simulations include the emissions and chemistry of nitrate-sulfate-ammonium aerosols coupled to gas-phase chemistry [45], carbonaceous aerosols [46,47], dust [39], and sea salt [40,41].

In this work, we used the 11-01 version of the GEOS-Chem model driven by GEOS-FP meteorology. We used the "tropchem" mechanism ("full chemistry" in the troposphere only) with secondary organic aerosols.

The layout of the Eastern European region considered for the study of aerosol spatiotemporal distribution is shown in Figure 1a. To avoid overloading the figures, the coordinates and city names will be omitted from further figures representing modeled aerosol distribution. The borders of the countries and seas in figures will assist to define the discussed areas.

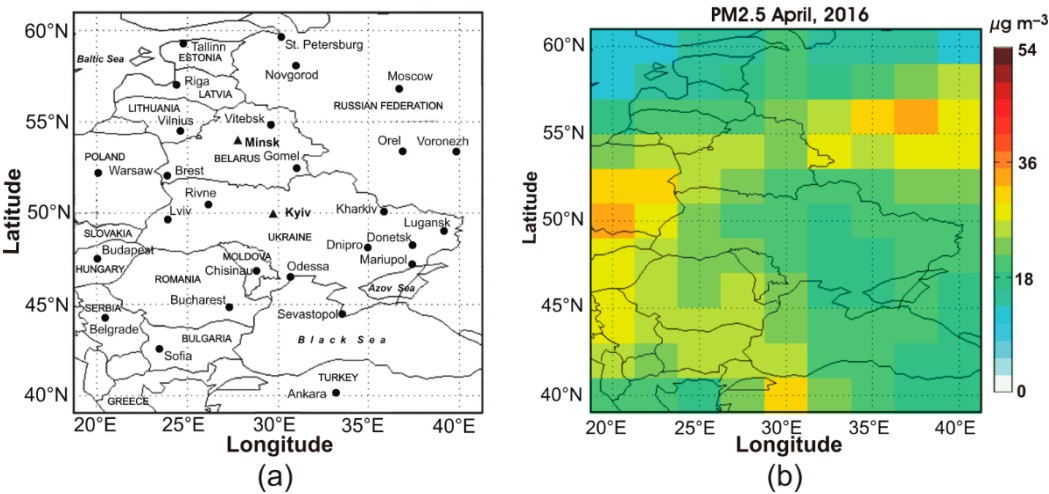

**Figure 1.** (**a**) The layout of the Eastern European region covered by modeling; (**b**) an example of the model distribution of the monthly averaged particulate matter (PM$_{2.5}$) concentrations in the atmosphere over the region in April 2016.

The simulation was performed on $2° \times 2.5°$ horizontal resolution latitude/longitude cells with 47 vertical layers up to 80 km. The cell center coordinates were at 40, 42, ..., 60° N; 20, 22.5, ..., 40° E (see Figure 1b). In total, the 99 cells covered the discussed Eastern European region, which includes the territories of Estonia, Latvia, Lithuania, Moldova, Romania, Bulgaria, Serbia, North Macedonia, Albania, as well as a substantial part of Poland, Slovakia, Hungary, Greece, Montenegro, Turkey, and Russia. The Azov Sea and the Black Sea were covered entirely, as well as part of the Baltic Sea. The calculations have been provided for the first vertical GEOS-Chem layer, from the surface up to about 100 m.

The data have been averaged monthly in 2016–2017 and are presented for each year. The PM$_{2.5}$, mineral dust, black carbon, organic aerosol, sea salt, nitrate, sulfate, and ammonium distributions are discussed. For each cell in Figures 1b, 2, 3, 4, 5, 6 and 7, the average concentration in micrograms per cubic meter (µg m$^{-3}$) is displayed through a color scale. For each aerosol type, the scale was chosen by accounting for the highest aerosol value, while the scale was different for each component and remained the same for one component for all 24 months. The seasonal behavior of the nitrate (NO$_3^-$), sulfate (SO$_4^{2-}$), and ammonium (NH$_4^+$) aerosols is considered, with an emphasis on the cold (January), warm (July), and transitional (March, November) months.

## 3. Results

### 3.1. PM$_{2.5}$

The particulate matter (PM$_{2.5}$) is defined as aerosol particles with an aerodynamic diameter of less than 2.5 µm. In GEOS-Chem, the PM$_{2.5}$ is considered as the sum of the nitrate, sulfate, ammonium, organic aerosol, black carbon, and sea salt in accumulation mode with an effective radius of 0.01–0.50 µm, mineral dust with an effective radius centered at 0.7 µm, and 38% of the mineral dust with an effective radius centered at 1.4 µm. The PM$_{2.5}$ calculation accounts for aerosol water using hygroscopic growth factors depending on the ambient relative humidity. The change in radius between the dry and wet aerosol is treated as a shell of water. Aerosol growth factors are computed using aerosol densities and aerosol radii.

The PM$_{2.5}$ concentration spatial distribution and seasonal variations from winter to autumn are presented in Figure 2. A seasonal enhancement of PM$_{2.5}$ content in wintertime was modeled from November to March in all areas. The most notable features were three areas with increased PM$_{2.5}$ concentration over the central and southern parts of Poland, Romania, and the area around Moscow. A significant difference was found between the PM$_{2.5}$ concentrations in March and July: 50–54 µg m$^{-3}$ vs. 15–18 µg m$^{-3}$.

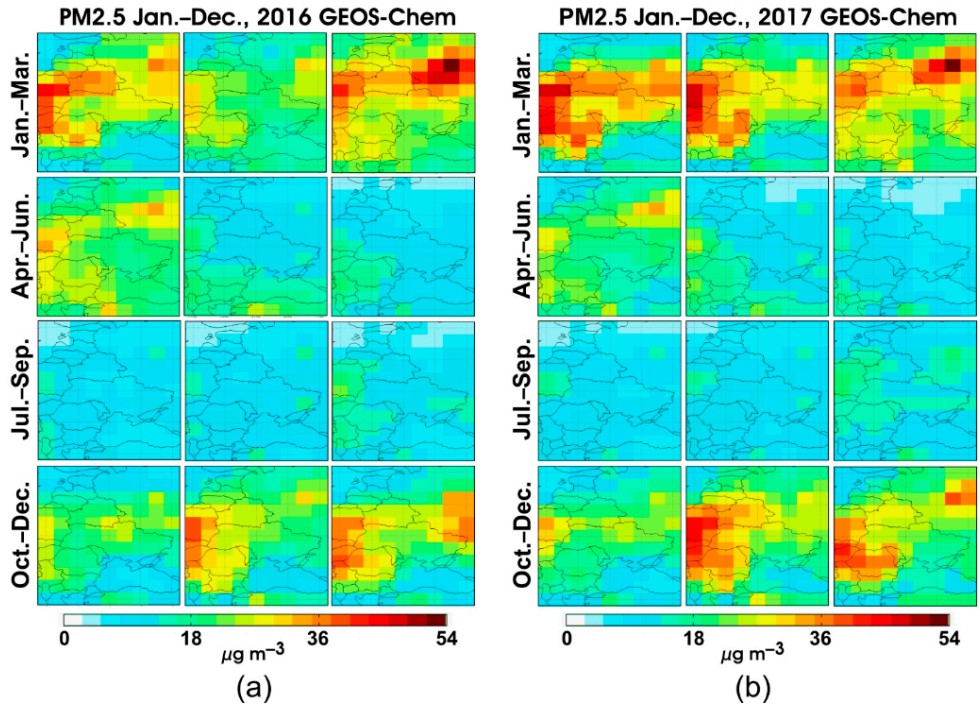

**Figure 2.** The spatial distribution of the monthly averaged PM$_{2.5}$ concentrations in µg m$^{-3}$ by the GEOS-Chem model calculation in (**a**) 2016 and (**b**) 2017.

In the central and southern parts of Poland, the highest PM$_{2.5}$ concentrations were found in January (40–50 µg m$^{-3}$). This PM$_{2.5}$ enveloped region was characterized by a large meridional extent, also covering Slovakia and Hungary in January 2016 and extending to Belarus, the northern Ukrainian region, and Romania in January 2017.

We noted that there was an exceptionally high contamination in March 2016 and 2017 over the area around Moscow City that exceeded the PM$_{2.5}$ concentration values in all considered regions and reached 54 µg m$^{-3}$. The cleanest atmosphere over the area, according to the model, was in the summer, when the PM$_{2.5}$ values were less than 15 µg m$^{-3}$, close to the threshold of contamination allowed by Directive 2008/50/EC on ambient air quality and cleaner air for Europe. A comparison of the PM$_{2.5}$ content showed systematically higher concentrations above Belarus compared to Ukraine. In January, the PM$_{2.5}$ values were 35 µg m$^{-3}$ and 25 µg m$^{-3}$ in 2016, and 38 µg m$^{-3}$ and 30 µg m$^{-3}$ in 2017, in Belarus and Ukraine, respectively.

A similar difference was found in March and April in both years. This can be explained by the larger influence of a high PM$_{2.5}$ aerosol contamination in Poland and Moscow on Belarus than on the territory of Ukraine. In summer, the PM$_{2.5}$ concentration is similar in Belarus and Ukraine.

*3.2. Mineral Dust*

Mineral dust contamination prevailed in the southern part of the territories considered here. Mineral dust mainly appeared in springtime in the southwest part of the area, with the highest concentration values in March and April 2016.

There were significant differences in the distribution and content of mineral dust between years (Figure 3a vs. Figure 3b). In 2017, the mineral dust concentration was much smaller in value and area than in 2016: 25 μg m$^{-3}$ vs. 40 μg m$^{-3}$ were the highest values.

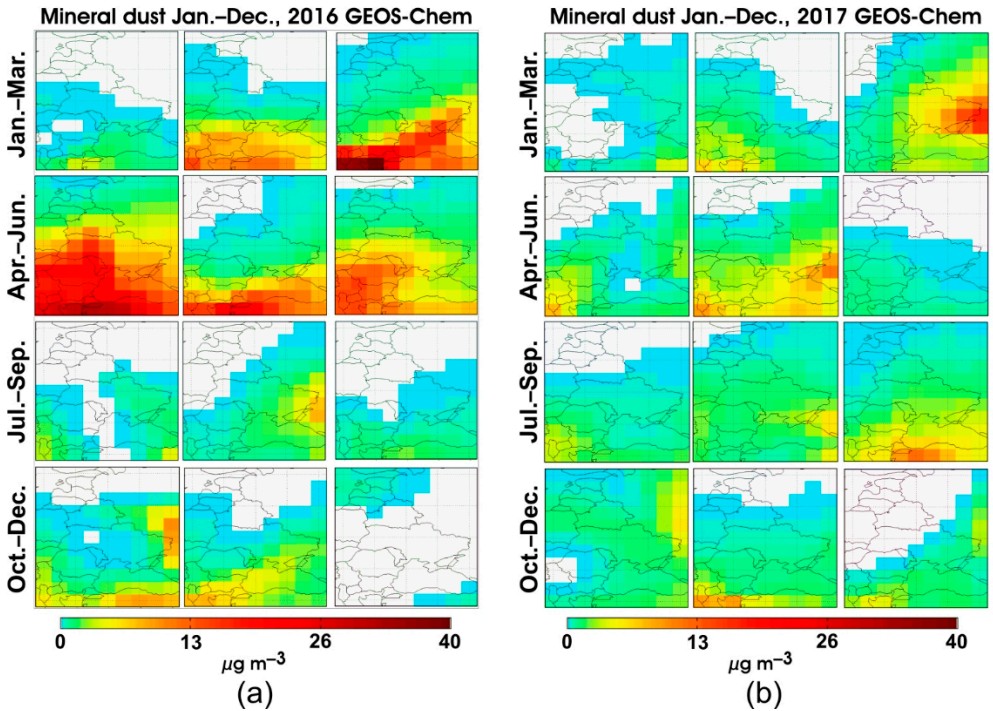

**Figure 3.** The monthly averaged mineral dust concentration's spatial distributions in (**a**) 2016 and (**b**) 2017.

### 3.3. Black Carbon

According to the model calculations, the black carbon (soot) content (Figure 4) in November–February significantly exceeded the content in April–September. The largest values were obtained over the industrial area in Southern Poland: 1.8 μg m$^{-3}$ in January and 1.0 μg m$^{-3}$ in July. The highest concentration of black carbon (BC) was seen over the Moscow area—in March, it was even larger than that in Southern Poland at that time, reaching 1.6 μg m$^{-3}$. The sustained soot concentrations from the surrounding areas were also modeled over Romania, but the corresponding values did not exceed 0.9 μg m$^{-3}$. The lowest black carbon content was calculated over the marine area in each of the seasons.

Over the territory of Ukraine and Belarus, the black carbon concentration was rather small, except in December and January, when the black carbon concentration increased to about 1.0 μg m$^{-3}$ in the western part of Ukraine, and to 0.8 μg m$^{-3}$ in the southwestern part of Belarus. The soot concentrations calculated for Ukraine and Belarus were significantly lower than the highest value in the region. Over Belarus, the soot content was systematically higher than it was over Ukraine in 2016. In January 2016, the largest values (over 1 μg m$^{-3}$) were simulated in the western areas of Belarus. The black carbon distribution was more homogeneous in March, with typical values of about 0.5 μg m$^{-3}$. The July values were not particularly different from those calculated for Ukraine, at about 0.3 μg m$^{-3}$. However, in 2017, the winter black carbon concentrations in the western parts of both countries were much higher than in 2016, reaching 1.3 μg m$^{-3}$. From Figure 4, we see that the black carbon content varied from year to year by 0.5 μg m$^{-3}$ in the winter months, and increased during a colder winter (2017, in comparison to 2016).

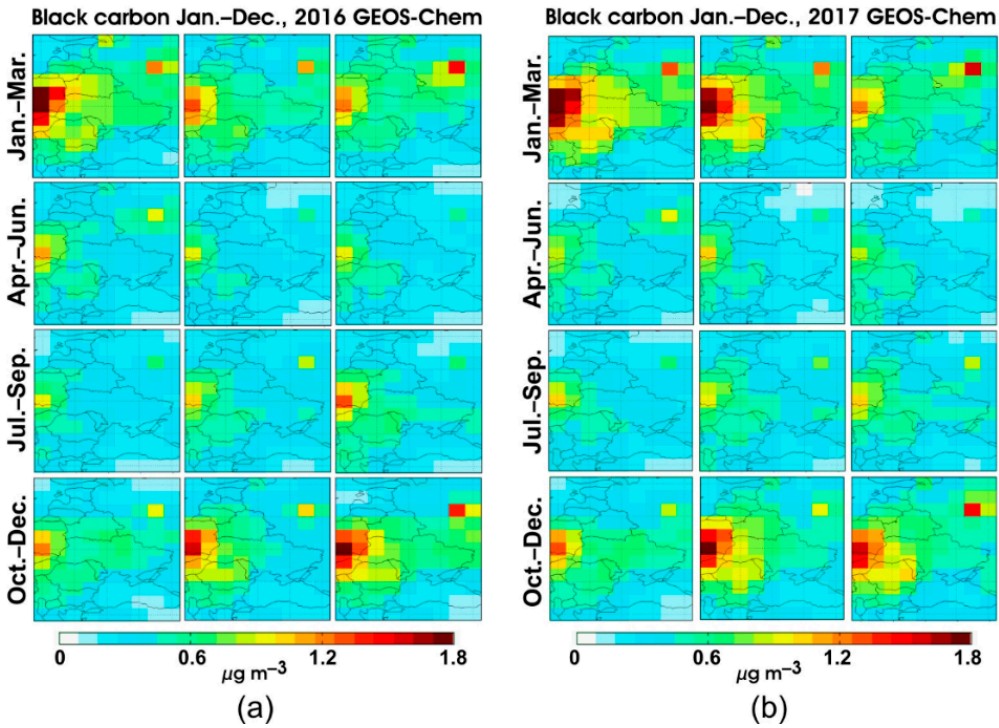

**Figure 4.** The black carbon monthly averaged concentrations in (**a**) 2016 and (**b**) 2017.

*3.4. Organic Aerosols*

The distribution of the monthly averaged concentration of organic aerosols, simulated with the GEOS-Chem model, is shown in Figure 5.

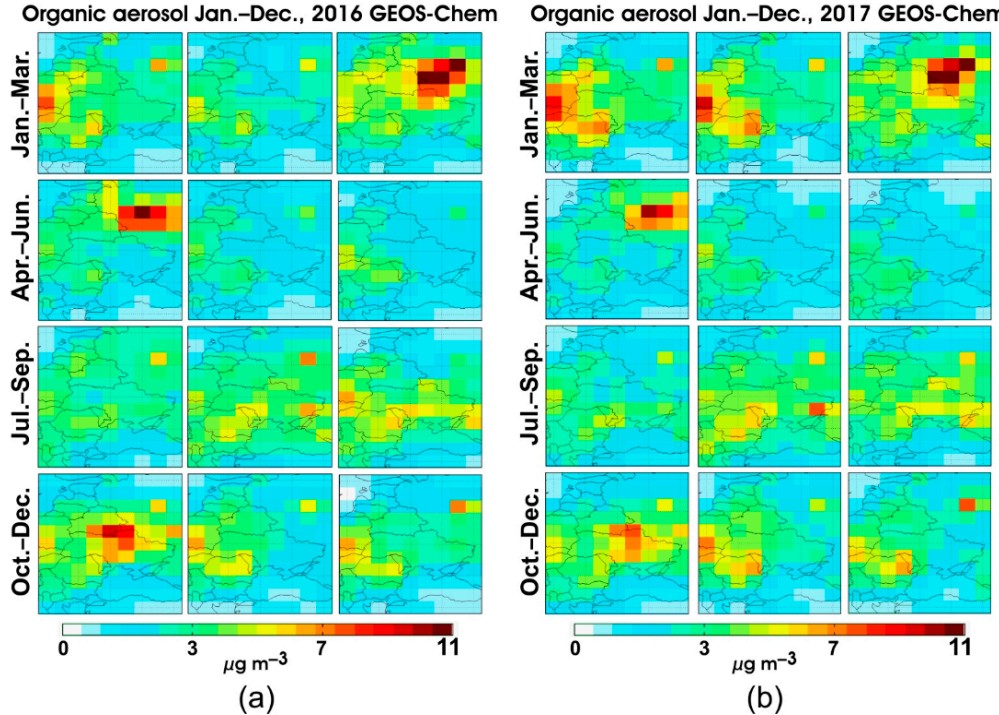

**Figure 5.** The spatial distributions of the monthly average concentrations of organic aerosols in (**a**) 2016 and (**b**) 2017.

In winter, a high organic aerosol concentration of 8 $\mu$g m$^{-3}$ was calculated over the southern part of Poland. A high organic aerosol concentration of 11 $\mu$g m$^{-3}$ was calculated over Moscow and regions to the west. High concentrations were found in March–April 2016, and for the same period of 2017. This March–April organic aerosol concentration increase was obtained by modeling for the large area that extends over the western regions of Russia, to the eastern border of Belarus and to the northern Ukraine border. An increased organic aerosol concentration of 7–9 $\mu$g m$^{-3}$ also appeared over Ukraine and Southeast Belarus in October in both years.

The seasonal behavior of the organic aerosol concentration in the whole area is complicated because the increased organic aerosol concentrations change their locations during the simulation. An increase in organic aerosol contamination was mainly located over the Moscow region in spring (March–April) and early autumn, with variations from year to year also revealed.

### 3.5. Sea Salt

The sea salt aerosol distribution was obviously connected to the main sources, which are marine areas (Figure 6). According to the model, the highest amounts of salt particles were located over the Baltic Sea and the Black Sea, which confirms the marine origin of this type of aerosol. The largest sea salt concentration of 7 $\mu$g m$^{-3}$ was over the southern part of the Baltic Sea in December, February, and March. The same sea salt concentration was obtained by a simulation over the northwest of the Black Sea in December and February.

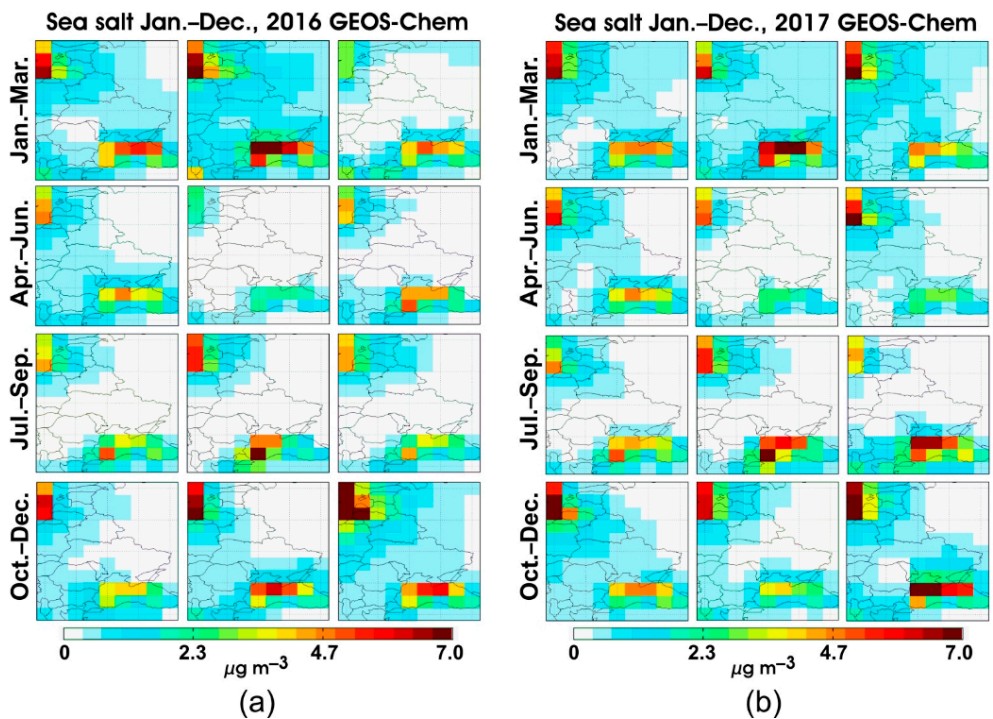

**Figure 6.** The spatial distributions of the monthly averaged concentration of sea salt in (**a**) 2016 and (**b**) 2017.

The spread of sea salt particles across land areas can be explained by the large area of the modeling cell and the aerosol transfer from the marine area, the latter of which depends on the wind strength and dominant direction. The seasonal sea salt aerosol content and distribution can be explained by the dominant factor in sea salt aerosol formation: the wind speed, which is stronger in the winter months (December–February (e.g., [48,49])).

### 3.6. Nitrate, Sulfate, and Ammonium Aerosols

The nitrate concentration's distribution (Figure 7a,b) displays a number of common features with the BC content (see Figure 4). An increased nitrate concentration was observed over a part of Poland and the Moscow region, where the values reached 6 µg m$^{-3}$ and 18 µg m$^{-3}$ in 2016 and 2017, respectively. As expected, the lowest nitrate concentration was calculated above the sea surface, similar to the BC distribution. The Ukraine territory was characterized by relatively low nitrate pollution, which, in particular, was less than over Belarus. An enhanced amount of nitrate appeared in November–March, with a very low concentration in May–September (see Figure 7a,b). The nitrate concentration exhibited clear seasonality; it differed between the cold and warm weather periods by more than 10-fold, from 1–2 µg m$^{-3}$ to 18 µg m$^{-3}$. In January, there was, almost continuously, a high-concentration band in the zonal direction at latitudes 51–55° N, which connects the Poland territory and the Moscow region, including the territory of Belarus. In March and November, in both years, this band became weaker, but the increased values over the Belarus territory remained.

The model of the sulfate spatial distribution in 2016 and 2017 is shown in Figure 7c,d. The levels of sulfate concentrations in the area ranged between 1 µg m$^{-3}$ and 6 µg m$^{-3}$, with mean values of 4 µg m$^{-3}$. This type of aerosol exhibits a strong seasonal increase in the spring months (March–April), with decreasing values from July to January. High sulfate concentrations were seen mostly over the southern and southwestern territories of the region. The increased amount of sulfate covered a large part of Romania, Bulgaria and Turkey (April–May), the eastern part of Ukraine (March–June), and a vast area around Moscow (March–April), reaching 4 µg m$^{-3}$. A prominent feature in the sulfate distribution was the high values over the west part of Turkey. This local increase of more than 6 µg m$^{-3}$ was seen during almost all months in 2016–2017.

In the atmosphere over the considered area, the ammonium concentration ranged from 0.3 µg m$^{-3}$ to 7 µg m$^{-3}$ (Figure 7e,f). The highest ammonium content was mostly found in the cold period of the year from November to March, especially March 2016 and January 2017. In March for both years, the highest concentration of ammonium was 7 µg m$^{-3}$ in the Moscow region. Similar to the seasonal variation in nitrate distribution, the ammonium concentration is low during the second half of the spring and summer months (May–September in 2016 and May–August in 2017).

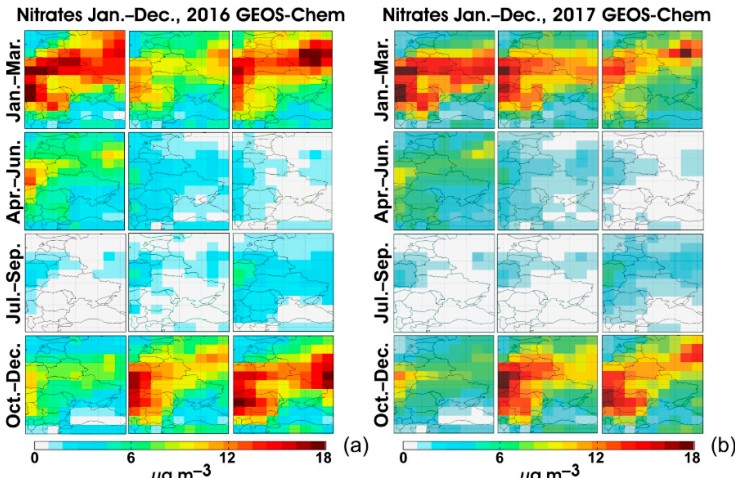

**Figure 7.** *Cont.*

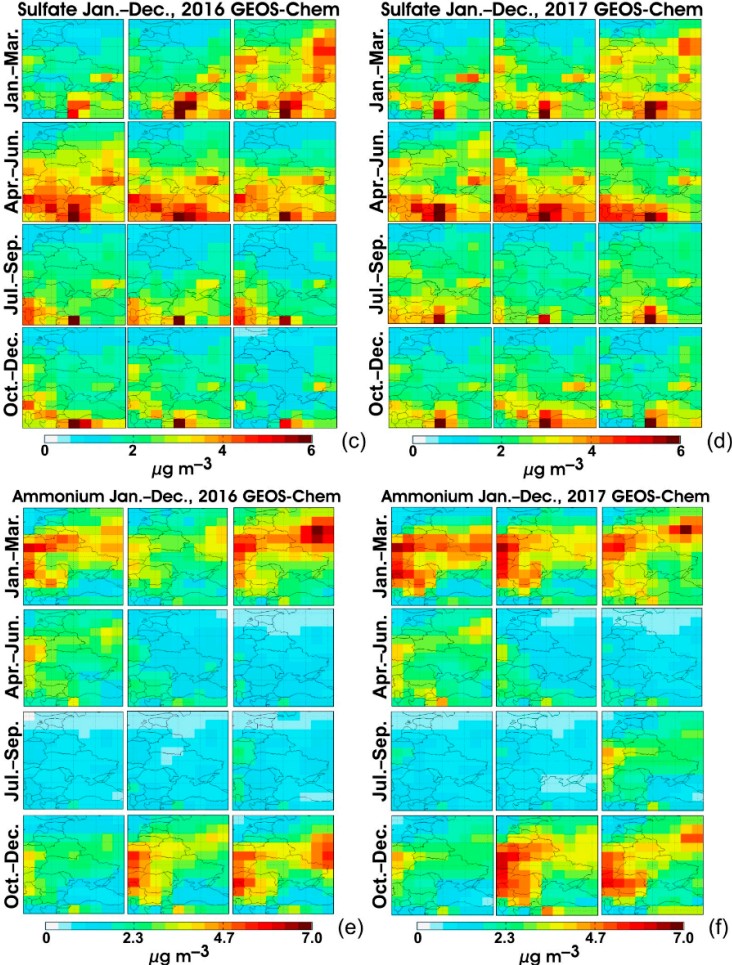

**Figure 7.** The spatial distributions of the monthly averaged (**a**,**b**) nitrate, (**c**,**d**) sulfate, and (**e**,**f**) ammonium concentrations (**a**,**c**,**e**) in 2016, and (**b**,**d**,**f**) in 2017.

## 4. Discussion

According to the GEOS-Chem model, the distribution of aerosols of various types in Eastern Europe has been analyzed. The model variations in the particulate matter $PM_{2.5}$ (which is a combination of the different aerosols), mineral dust, black carbon (soot), organic aerosols, and sea salt particles were considered in detail. The behavior of the concentration of the nitrate, sulfate, and ammonium aerosols was also assessed. The seasonal changes in the aerosol constituent concentration, based on the results of a two-year model, were discussed with a focus on the spatial distribution and seasonal variations.

*Particulate matter $PM_{2.5}$*. The lowest $PM_{2.5}$ values were modeled over the sea. This was especially noticeable for November and January, when the value over the sea was 10 $\mu g\ m^{-3}$—three times less than over the land. At the same time, in January, a small $PM_{2.5}$ concentration was calculated over the territory of Turkey. The $PM_{2.5}$ values in March and July were more homogeneous over the sea and above the surrounding land, however, at different levels: the $PM_{2.5}$ was about 20 $\mu g\ m^{-3}$ in March and about 5 $\mu g\ m^{-3}$ in July.

A high $PM_{2.5}$ concentration was modeled in January 2016 and December 2016–February 2017 (40–50 $\mu g\ m^{-3}$) over the central and southern parts of Poland. This corresponds to measurements of when indoor heating is used, as presented in Sówka et al. [50]. There is a clear seasonal cycle in the largest $PM_{2.5}$ distributions (Figure 2) that can be associated with the heating period when coal is used for heating houses in those areas. Increased $PM_{2.5}$ values are retained for territories with high anthropogenic loads: the South of Poland and the Moscow area. The $PM_{2.5}$ content was larger in winter 2017, especially in January–February (e.g., [51]).

The highest PM$_{2.5}$ concentration of 54 µg m$^{-3}$ was calculated in March, which is the only month during both 2016 and 2017 when the magnitude of the PM$_{2.5}$ over the Moscow area was greater than that over Poland. In January and November, the highest PM$_{2.5}$ value was 1.5 times smaller than in March, and the size of the PM$_{2.5}$ enhanced concentration area was also smaller, especially in the fall.

*Mineral dust.* Mineral dust aerosols represent soil particles suspended in the atmosphere. The comparison of the PM$_{2.5}$ and mineral dust by distribution and content variations (see Figures 2 and 3) reveals a significant difference. According to the modeled data, the mineral dust concentration increases over the southern part of the area, while the PM$_{2.5}$ mostly impacts the western and northeastern areas. This result is confirmed by [52], where the main source of mineral dust is from the Saharan Desert via the Mediterranean Sea. The mineral dust content increased in springtime in the southwestern part of the area, whereas the PM$_{2.5}$ mostly increased in the winter months. The significant difference in the mineral dust distribution and content from year to year is explained by the difference in atmospheric circulation and Saharan dust outbreak in spring 2016, when a large amount of mineral dust was transported from North Africa to Europe. The mineral dust enhancement in the eastern part of Ukraine was caused by wind erosion in the steppe region in March and May 2017, similar to the strong dust storm in 2007 [53].

*Black carbon.* Black carbon is a constituent of particulate matter that results from the incomplete combustion of coal and other fossil fuels, vehicle engine operation, and biomass-burning. The enhanced concentration of inorganic carbon is mostly related to the production of energy at thermal power plants and the heating of houses, especially if coal is used. Although black carbon comprises a small part of atmospheric aerosols over Europe (less than 10% of the PM$_{2.5}$ if we compare Figures 2 and 4), it might cause a significant increase in radiative forcing (e.g., [54]). A seasonal cycle with a peak in the cold-weather period, when there is a need for heating the house, is seen over Central and Eastern Europe [50,55,56].

Our model calculations show that the black carbon content in November–February (the heating season) significantly exceeds the content in April–September (the non-heating season). The high values were simulated over industrial areas in Southern Poland: 1.8 µg m$^{-3}$ in January and 1.0 µg m$^{-3}$ in July, which is similar to the elemental carbon data in [50]. The black carbon concentration over the Moscow area was at its largest in March, amounting to about 1.6 µg m$^{-3}$, which is close to the values measured during the AeroRadCity-2018 spring aerosol experiment in Moscow [57]. These comparisons confirm the reliability of the GEOS-Chem evaluations of black carbon content.

*Secondary organic aerosols.* The organic component of aerosol particles is a complex mixture of hundreds of organic compounds. Secondary organic aerosols, formed in the atmosphere by the oxidation of organic gases, represents a major fraction of global submicron-sized atmospheric organic aerosols [58]. According to our simulation, the organic aerosol concentration is particularly high over the polluted urban regions of Eastern Europe, such as the western regions of Russia, the Moscow region, the South of Poland, and the central part of Ukraine, in late autumn, winter and early spring. The largest organic aerosol concentrations were over 11 µg m$^{-3}$, which is comparable with the results of [50], showing organic aerosol (in particular, organic carbon) concentrations over Poland's cities of about 5 µg m$^{-3}$ in the non-heating season and 17 µg m$^{-3}$ in the heating season. The increase in the organic aerosol concentration also corresponds to the large black carbon and PM$_{2.5}$ concentrations over the Moscow area in March in both years, according to our data. These features of the organic aerosol distribution suggest that anthropogenic sources are related to fuel-burning products in the regions under study. The simulated organic aerosol distribution also correlates to the PM distribution over the region under study during the cold season, because organic aerosols are involved in PM. The results of our simulation of the organic aerosol spatial distribution and level of concentrations correspond to their common features described above. Our results also suggest a mainly anthropogenic genesis of organic aerosol sources in the regions under study, but their influence on the air quality and climate changes have to be studied more carefully, using both model simulation and remote and in situ observations.

*Sea salt.* Sea salt aerosols are introduced into the atmosphere as primary particles of natural origin, mainly by sea spray. Sea salt particles represent about 30% of the global column aerosol optical depth. In many areas, the sea salt aerosol concentration is close to that of water vapor, corresponding to the wet scavenging dominant role of sea salt aerosol removal from the atmosphere. However, wet scavenging in a drier and colder winter atmosphere is less than in a wetter summer atmosphere ([49] and citations therein), so proper quantification of the sea salt aerosol content is an important task for a better implementation of sea salt aerosols in climate models [48].

We have supposed that the sea salt aerosol has a predominantly (and perhaps almost exclusively) marine origin. There are two distinct areas with the highest sea salt concentration in the studied territory: the Black Sea and the Baltic Sea. A systematic increase was also noticeable for cells that included a part of the Mediterranean Sea or its branches. However, in January–March and in October and December, a sea salt aerosol concentration of about 1 µg m$^{-3}$ was found in most regions under study. This can be considered a result of the sea salt aerosols' wind transport from both seas.

*Nitrate, sulfate, and ammonium.* Nitrate-sulfate-ammonium aerosols are present in the atmosphere as solid particles, or droplets of water solutions including sulfuric and nitric acid, partially or totally neutralized by ammonia. Sulfate and nitrate aerosols may be present in the atmosphere, and are associated with sodium, or calcium, or other metals. A small fraction of the sulfate aerosol is emitted as sea salt primary particles. Most nitrate-sulfate-ammonium aerosols are formed in the atmosphere through the oxidation and neutralization of precursor gases: sulfur dioxide ($SO_2$), nitric oxides ($NO_x$), and ammonia ($NH_3$) [45]. The major anthropogenic source of $SO_2$ and $NO_2$ is fossil fuel combustion. Important natural sources of $NO_2$ include lightning, soils, and wildfire. Major sources of ammonia emissions are animals, fertilizers, industry, fossil fuels, biomass-burning, oceans, crops, and soils [59]. As precursor gases generally have a lifetime of about one day, the distribution of nitrate-sulfate-ammonium aerosols corresponds to the distribution of sources of sulfur dioxide, nitric oxides, and ammonium [45,59].

According to our calculations, the highest nitrate concentration values are in the range of 6–18 µg m$^{-3}$, depending on the year. The modeled ammonium concentration varies within the range of 0.3–7 µg m$^{-3}$ over Eastern Europe, with increased values in the autumn, winter, and spring in the eastern part of the area and in the Moscow region (Figure 7e,f). The nitrate and ammonium spatial distributions and seasonal behaviors are similar in general (Figure 7a,b,e,f). The model concentration of these aerosol species exhibits clear seasonal variations, with increasing values in November–March. The GEOS-Chem model simulations of the nitrate-sulfate-ammonium aerosols, as well as comparisons to observations in Europe, have been provided by Park et al. [45]. The comparison of model calculations and the European Monitoring and Evaluation Programme observations indicate the determination coefficient $R^2 = 0.48$, 0.63, and 0.72 for nitrate, ammonium, and sulfate, respectively [45]. Ammonia in the atmosphere originates predominantly from the agricultural industry, fossil fuels, fires, and biomass-burning [60]. Ambient ammonia concentrations change over several orders of magnitude, which complicates comparisons between measurements and models. The comparison of the GEOS-Chem model data with both remote and in situ measurements was performed in [60,61]. Schiferl et al. [61] concluded that the GEOS-Chem simulation underestimates the ammonia concentration near the main sources by 26%. Zhu et al. [62] implemented into GEOS-Chem the bidirectional exchange of ammonium, considering the equilibrium between ammonia ($NH_3$) and ammonium ($NH_4^+$). They emphasized that the GEOS-Chem model typically underestimates ammonium concentrations, sometimes by 2- or 5-fold, in the spring and autumn.

The monthly variations in the sulfate concentration (Figure 7c,d) show the highest seasonal values in spring and early summer, unlike nitrate and ammonium. The sulfate spatial distribution also demonstrates differences from those of nitrate and ammonium, with the highest sulfate concentration over the southern and southwestern areas of Eastern Europe, similar to the distribution modeled by Yang et al. [63]. The area of increased sulfate concentration is seen repeatedly every month (Figure 7c,d).

This feature is located in the western part of Turkey and requires more detailed consideration in future work.

The analysis of the seasonal variations and distribution of aerosol concentration shows some similarities between the different species. There are some specific features described above, but in general the highest aerosol concentration is simulated over the Poland and Moscow region. In most cases, the model demonstrates, in particular, higher concentrations in Belarus than in Ukraine. The reliability of such a conclusion requires verification based on direct observations by a sun photometer, LiDAR, and in situ measurements. For most aerosol species—$PM_{2.5}$, black carbon, and organic aerosols— a clear seasonal cycle exists, with the largest values found in the colder period of the year.

## 5. Conclusions

The content of this paper is a simulation by the GEOS-Chem model and a consideration of the monthly averaged properties of the aerosols' distribution over Eastern Europe in the near-surface layer below 100 m for 2016–2017. The GEOS-Chem v11-01 version model, driven by GEOS-FP meteorology with the "tropchem" mechanism, was used. The areas under consideration also include the Azov Sea, the Black Sea, and the southern part of the Baltic Sea. The $PM_{2.5}$, black carbon, mineral dust, organic aerosol, sea salt, nitrate, sulfate, and ammonium distributions were discussed. Comparisons of the GEOS-Chem model data with observations at various altitudes in the atmosphere have been undertaken by other authors [45,59,62,63]. These comparisons show a reasonable agreement between the model and the observed data, which in general justifies the results of this paper.

An increased $PM_{2.5}$ concentration corresponds to urban areas, so this type of aerosol is likely of mostly anthropogenic origin. The mineral dust aerosols are not distributed like $PM_{2.5}$, and are composed of dust hypothetically transported by the wind from the Sahara Desert and the Ukrainian steppe. The content of black carbon in the atmosphere is mostly related to the production of energy by thermal power plants and the heating of houses, which use coal. Our simulations show that the black carbon content in November–February significantly exceeded that of April–September, essentially over urban territories, similar to the $PM_{2.5}$ distribution.

According to our simulation, the organic aerosol concentration is substantially higher over polluted urban regions of Eastern Europe and corresponds to a high concentration of black carbon and $PM_{2.5}$. These results suggest a mainly anthropogenic origin of organic aerosols in Eastern Europe. Sea salt aerosols are introduced into the atmosphere as primary particles of natural origin by sea spray. The sea salt aerosols' concentration reaches the highest values over the Black Sea and the Baltic Sea. An almost negligible concentration of sea salt aerosols over the land under consideration can be considered a result of the wind transport of sea salt aerosols from both seas.

GEOS-Chem modeling provided a general view of concentration levels and seasonal aerosol variations, which are on average consistent with the measurements and models in other regions (see e.g., [62,63]). The results of the GEOS-Chem aerosol properties simulation in our region allowed us to understand the needs and extent of development of the comprehensive air-quality network in countries that are sparsely covered by aerosol measurements. Therefore, GEOS-Chem modeling makes it possible to create a baseline of the studied aerosol spatial distributions and to determine the areas and species that have to be investigated in more detail.

**Author Contributions:** Conceptualization, G.M. and N.M.; Formal analysis, A.G. and V.D.; Funding acquisition, A.L., V.K., and Y.S.; Investigation, Y.Y. and Y.W.; Methodology, N.M. and A.G.; Project administration, G.M. and N.M.; Software, N.M.; Supervision, G.M.; Validation, G.M. and V.D.; Writing—original draft, G.M., N.M., and V.D.; Writing—review and editing, G.M., N.M., V.D., A.B., A.C., Y.Y., Y.W., A.L., V.K., and O.T. All authors wrote parts of the text, contributed to the discussion of the results, helped edit the manuscript. All authors have read and agreed to the published version of the manuscript.

**Funding:** This research received no external funding.

**Acknowledgments:** This work was supported in part by the College of Physics, International Center of Future Science, Jilin University, China, and by Taras Shevchenko National University of Kyiv, projects 19BF051-08 and 20BF051-02. The authors would also like to acknowledge the NATO project No. G5500 and the

SMURBS/ERA-PLANET/GEO Essential project supported by the European Commission "Horizon 2020" program. The authors thank the two anonymous referees for their useful comments and suggestions.

**Conflicts of Interest:** The authors declare no conflict of interest.

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
