# Peer review of "Atmospheric Aerosol Distribution in 2016–2017 over the Eastern European Region Based on the GEOS-Chem Model"

_atmosphere, doi:10.3390/atmos11070722_

Round 1

Reviewer 1 Report

The manuscript ‘Atmospheric Aerosol Distribution in 2016–2017 over the Eastern European Region Based on GEOS-Chem Model’ by Milinevsky et al. used a GEOS-Chem model to study the concentration of different aerosol components distributed over the Eastern European region for the years 2016 and 2017.

Based on their simulations, the authors present the data for different periods of 2016 and 2017 for different aerosol components: PM2.5 concentration, mineral dust, black carbon, organic aerosol, and sea salt. The authors reported seasonal and spatial variation of these aerosol components simulated over the Eastern European region.

Overall, the study is interesting in providing useful information on the concentration of atmospheric aerosol from different sources for the Eastern European region. However, these seasonal and spatial values for 2 consecutive years are the simulated values. The authors need to validate their values for each aerosol component with the measured values in the region and need to report how the simulated values compare/differ with the measured values. The comparison between the measured and simulated values should be as explicit as possible. If the measurements for all simulated components are not available, it must be mentioned.

I am not clear whether the authors mean the maxima or maximum while reporting the aerosol concentration. Maxima and minima are not the same as maximum and minimum.

In many places, the authors mentioned that the values are significantly different/higher/lower. But I did not see any statistical analyses, statistical tests used, and the level of significance. The use of the terms like ‘almost negligibly small’ is vague and authors have to be specific in making their statement.

I have a serious concern about the English language used. Though English is not my native language, I found several flaws in the language throughout the manuscript. Some sentences require to be rephrased to communicate the message. In some sentences, the use of article is not correct while prepositions are missing in some sentences. Authors must pay attention to properly communicate their findings to the readers. The English language needs to be improved and polished. Both the abstract and conclusion suffer from language problems.

GENERAL/SPECIFIC COMMENTS

  1. It is ‘lifetime’ not ‘time-life’. See line 51.
  2. Aerosol ‘concentration’ and aerosol ‘contamination’ are used interchangeably. Based on the simulation and unit, it is aerosol concentration not aerosol contamination. Correct it and be consistent.
  3. In the introduction section you mentioned that section 4 is your conclusions section. But you have your conclusions in section 5. I suggest having a result and discussion together in the same section. One has to go back to the result section to look at data when you have a separate section for discussion. Also, present only critical data that are relevant for discussion in your result. Readers can always infer to the Figures to look at values for different time frames. Authors can report maximum/minimum values for months/season in a Table in the supplementary material if they want.
  4. On lines 112-113, you mentioned that ‘The GEOS-Chem is a global three-dimensional chemical transport model, which is the convenient instrument for evaluation..’. It gives the impression that you are using an instrument which is not the case. Replace the term ‘instrument’ with a suitable word.
  5. The term ‘heating season’ in the manuscript can be ambiguous. Please define the term clearly. I suggest to define and use ‘heat period’ instead of the ‘heating season’. Also define what you mean by the ‘cold period’.
  6. Be more explicit when you compare your simulated values from the model with the measurement values for each aerosol components to prove the robustness and reliability of the model you used.
  7. I know the citations for this journal are numeric. But when you can name the first author et al., use the first author et al. [citation #] format, where possible. For example in ‘The seasonal aerosol properties in the East-European region were investigated in [9].’ can be written as ‘The seasonal aerosol properties in the East-European region were investigated in Milinevsky et al. [9].’

Author Response

Dear Reviewer 1

We thank for your useful comments and proposed corrections. We revised the manuscript according your proposals as much as possible. See our corrections in attached file and corrections in the manuscript text in blue color. 

Reviewer 2 Report

This interesting paper presents the temporal and spatial variability of the PM2.5 content for the different families of particles over eastern Europe, using the GEOS-Chem model. The paper is well written, and relevant references are given. The main problem is that the results are coming only from modelling calculations. The results are provided with an accuracy of 0.1µg and no error bars are given. How can we be sure that these values are realistic? There is a need of comparison with some in-situ measurements (typically from air-quality networks) obtained at the same time to validate the modelling results for total mass concentrations and for all the particles families, as partly done in lines 297-300 for black carbon.

Also, here are some specific comments:

Line 161: Why putting the nitrate, sulfate and ammonium in a supplementary material? These results are of interest and can be included in the main paper.

Line 167-169: What are the origins of these values (effective radius, %)? Are they input parameters for the model, or results of the simulations?

Line 248-250 are strange. They look like a reviewer comment.

Considering these comments, I think the paper could be published in a revised version.

Author Response

Dear Reviewer 2

On behalf of authors, I thank your for useful comments and proposed corrections. We revised the manuscript according your proposals. See our corrections and comments in attached file and corrections in the manuscript text in blue color. 

Round 2

Reviewer 1 Report

English language must be improved and polished. The authors need to go through this once again and need corrections for the technical part of the English language in the manuscript.

I am aware that 'maxima' and 'minima' are plural forms for 'maximum' and 'minimum', respectively. It is known to us all that 'maxima' and 'minima' are typically used for mathematical function and here the terms are not meant for the mathematical function. In that case, I would always use the terms like 'highest', 'lowest' to quantify aerosol values.  

The authors clarified that they have no ground-based data for aerosol components simulated in the model and added in the conclusion section that they want to create a baseline for the distributions of studied aerosols and to determine the areas and aerosol species for detail ground-based study in the future. I suggest the authors make the purpose of this paper clear at the end of the introduction section as well. In the absence of comparison with in-situ measurements and statistical analyses, the model results thus lack validation and this will be a weaker aspect of this paper.

Author Response

We thank Reviewer for useful comments and corrections. We corrected the text and included new one (in blue color) according Reviewer suggestions. Our revisions are given below line-by-line.

Reviewer #1:

Comments and Suggestions for Authors

Reviewer Comment: English language must be improved and polished. The authors need to go through this once again and need corrections for the technical part of the English language in the manuscript.

Author Comment: We look through the text once again. We also provided the MDPI English Editing Service for the text. Hopefully that improved English of the manuscript.

RC: I am aware that 'maxima' and 'minima' are plural forms for 'maximum' and 'minimum', respectively. It is known to us all that 'maxima' and 'minima' are typically used for mathematical function and here the terms are not meant for the mathematical function. In that case, I would always use the terms like 'highest', 'lowest' to quantify aerosol values.

AC: We made corrections in all text (line numbers from old text):

Line 202, L205, L209: "maximum" corrected to "highest"

L218: old text "lower than the regional maximum" corrected to "lower than the highest value in the region"

L243: corrected to "highest values"

L257: corrected to "lowest concentration was"

L269: "seasonal maximum" corrected to "seasonal growth"

L275: "local maximum" corrected to "local increase"

L279: "the maximum of ammonium concentration" corrected to "the highest concentration of ammonium"

L363: "two distinct sea salt maxima" corrected to "two distinct areas with highest sea salt concentration"

L400-L401: "represent a seasonal maximum " corrected to "represent the highest seasonal values"

L409: "aerosol concentration value maxima" corrected to "the highest aerosol concentration"

L436: "demonstrates maximum" corrected to "reaches the highest values"

RC: The authors clarified that they have no ground-based data for aerosol components simulated in the model and added in the conclusion section that they want to create a baseline for the distributions of studied aerosols and to determine the areas and aerosol species for detail ground-based study in the future. I suggest the authors make the purpose of this paper clear at the end of the introduction section as well.

AC: The old text in L104-110

"There are no detailed measurements of the aerosols concentrations and properties in the ground air and in the atmosphere boundary layer over Belarus and Ukraine. Our research is the first attempt to estimate the possible content of aerosols of various compositions in the surface air based on the GEOS-Chem model in the region. The purpose of this paper is to simulate and consider the monthly variation properties of the atmosphere aerosols over the Eastern European region with special focus on Belarus and Ukraine countries, where ground-based aerosol observations by species are not available."

Replaced by text in L104-108 according Reviewer suggestion:

"The main purpose of this paper is to determine typical aerosols and to reveal distinctive features of their spatio-temporal distributions and variations in the atmosphere over Belarus and Ukraine where detail in-situ aerosol observations are not available. We use the GEOS-Chem model to simulate variation of aerosols monthly properties in the atmosphere over the Eastern European region with special focus on Belarus and Ukraine. "

AC: The old text in L439-444

The main purpose of the paper is to create a baseline of the studied aerosol spatial distributions and to determine the areas and species which have to be investigated in more detail. This result allows us to understand the needs and extent of development of the comprehensive air-quality network in countries which are sparsely covered by aerosol measurements. The GEOS-Chem modeling provides a general view of concentration levels and seasonal aerosol variations, which are in average consistent with other measurements and models (see e.g., [62,63]).

Replaced by text in L438-44 for consistency

The GEOS-Chem modeling provided a general view of concentration levels and seasonal aerosol variations, which are in average consistent with measurements and  models  in other regions (see e.g., [62,63]). Results of the GEOS-Chem aerosols properties simulation in our region allows us to understand the needs and extent of development of the comprehensive air-quality network in countries which are sparsely covered by aerosol measurements. Thereby the GEOS-Chem modeling makes it possible to create a baseline of the studied aerosol spatial distributions and to determine the areas and species which have to be investigated in more detail.